# Molecular Dynamics Simulation of Polymer Nanocomposites with Supramolecular Network Constructed via Functionalized Polymer End-Grafted Nanoparticles

**DOI:** 10.3390/polym15153259

**Published:** 2023-07-31

**Authors:** Guanyi Hou, Runhan Ren, Wei Shang, Yunxuan Weng, Jun Liu

**Affiliations:** 1College of Chemistry and Materials Engineering, Beijing Technology and Business University, Beijing 100048, China; renrunhan@163.com (R.R.); herzflut@foxmail.com (W.S.); 2Center of Advanced Elastomer Materials, Beijing University of Chemical Technology, Beijing 100029, China; lj200321039@163.com

**Keywords:** self-healing materials, supramolecular crosslink network, molecular dynamic simulation

## Abstract

Since the proposal of self-healing materials, numerous researchers have focused on exploring their potential applications in flexible sensors, bionic robots, satellites, etc. However, there have been few studies on the relationship between the morphology of the dynamic crosslink network and the comprehensive properties of self-healing polymer nanocomposites (PNCs). In this study, we designed a series of modified nanoparticles with different sphericity (η) to establish a supramolecular network, which provide the self-healing ability to PNCs. We analyzed the relationship between the morphology of the supramolecular network and the mechanical performance and self-healing behavior. We observed that as η increased, the distribution of the supramolecular network became more uniform in most cases. Examination of the segment dynamics of polymer chains showed that the completeness of the supramolecular network significantly hindered the mobility of polymer matrix chains. The mechanical performance and self-healing behavior of the PNCs showed that the supramolecular network mainly contributed to the mechanical performance, while the self-healing efficiency was dominated by the variation of η. We observed that appropriate grafting density is the proper way to effectively enhance the mechanical and self-healing performance of PNCs. This study provides a unique guideline for designing and fabricating self-healing PNCs with modified Nanoparticles (NPs).

## 1. Introduction

As one of the most significant soft materials, polymer nanocomposites (PNCs) exhibit remarkable comprehensive properties such as static elastic modulus, tensile strength, flexibility, abrasion performance, tear resistance, etc. By introducing nanoparticles and crosslinking networks [1], these unique performances make PNCs widely employed in numerous applications, especially for tire tread materials [2,3,4]. However, traditional PNCs are unfixable when damaged, and when they are no longer useful, it is challenging to recycle the materials into feedstock. Therefore, many scientists are exploring ways to endow PNCs with self-healing abilities to address these challenges [5,6,7,8].

There are typically two types of self-healing materials: extrinsic healing systems and intrinsic self-healing materials. Intrinsic self-healing materials are usually based on non-covalent chemistries or dynamic covalent chemistries [9]. For instance, Perera introduced a dynamic covalent adaptive network into hydrogel, designing a novel soft-material with self-healing, shape memory, and stimuli-induced stiffness changes [10]. Liu, et al., inserted sacrificial bond into covalent network to enhance the dynamics mechanical performance of PNCs. The highlight of this research is that the distribution of sacrificial bonds accelerates the rearrangement of network topology networks, significantly enhancing the overall mechanical properties of PNCs [11]. However, due to complex influencing factors, such as the fraction of nanoparticles (NPs), interfacial compatibility between matrix and fillers, the mechanism by which nanofillers affect the conformation of covalent networks is not fully understood, and the relationship between the distribution of the conformation of physical crosslinking networks is unclear.

To discover the influence of nanofiller on the distribution of dynamic crosslinking network, numerous studies have been conducted [12,13,14,15,16,17,18]. An, et al., discovered that the moderate fraction of modified boron nitride would not affect the distribution of dynamic crosslinking network, which would significantly improve the tensile strength and the elongation of PNCs [19]. Wang, et al., found that when the weight fraction of modified silica NPs was approximately equal to 1%, the mechanical performance of hydrogel was maximized [20]. Park, et al., demonstrated that the introduction of graphene-oxide will stimulate the bond exchange reaction, significantly reinforcing the self-healing efficiency of PNCs [21]. Recently, the use of modified NPs as link-points to establish dynamic crosslinking networks has garnered attention from scientists [22,23,24,25]. Barrios, et al., combined sulfur (S) and zinc oxide (ZnO) as vulcanizing agents in a carboxylated nitrile rubber (XNBR) matrix to establish a dual network aimed at improving the material’s abrasion resistance. The mechanical performance of XNBR was significantly improved (up to 26%) [26].

Computer simulations offer several advantages for investigating the static and dynamic properties of polymer networks over traditional experimental approaches, including better control of network formation and precise knowledge of dynamic network features [27]. For instance, Wick et al., employed the molecular dynamics simulation (MD) method to create several dual-crosslinking systems and analyze which dynamic crosslinking density can endow the system with the greatest mechanical performance. The results indicated that at higher crosslinking densities, the material exhibited excellent self-healing performance and high tensile strain [28]. Fu et al., combined MD and experimental methods to investigate the thermal properties and microstructure morphology of A (DGEBA)/methyl tetrahydrophthalic anhydride (MTHPA) and DGEBA/nadic anhydride (NA). The experimental results confirmed the accuracy of the crosslinking model and indicated that the slight change in the curing agent structure significantly affected the synergy rotational energy barrier, cohesive energy density, and free volume fraction, thus affecting the glass transition temperature (T_g_) and modulus of the system [29]. However, few studies have focused on the influence of NPs on the formation of dynamic crosslink networks and the performance of PNCs.

Therefore, in this study, the coarse-grained molecular dynamics simulation (CGMD) method is employed to investigate the influence of nanoparticle (NP) structure on the formation of a dual network (chemical/supramolecular crosslink network). Various modified NPs with different structures are designed and introduced into a polymer matrix with a dynamic crosslink network. Taking the previous works as the reference [16,17,30,31,32], this supramolecular dynamic network is further constructed via the strong interaction between the end-groups of the grafted nanoparticles and grafted polymer chains. The influence of the completeness of the supramolecular network on the mechanical properties and self-healing efficiency of the corresponding composite materials is further studied, where the completeness is defined as the supramolecular cross-linked network structure is fully and uniformly represented throughout the system, indicating that there are no areas of the system that lack the cross-linked network or have disproportionately large clusters of the network. This term implies that the network is evenly distributed and integrated throughout the entire system. This research is expected to provide guidance for the design and fabrication of self-healing polymeric materials.

## 2. Simulation Methods

In this study, the coarse-grain MD simulation were used to study the effect of the structure of the NPs. For this simulation model, different number of short chains (*L* = 4σ, composed with 4 beads with diameter, *D* = 1σ) with modified end group are grafted on the core NP (*D* = 4σ) (Figure 1). In our previous research, we proposed a concept named “sphericity (η)”, in which the number of arms was represented by divisions of “circle” [33]. In this study, we use this idea to definite the grafted density of modified-NPs, where η = 0 represents that there is no grafted polymer chain attached on the core-NP, and η = 1.0 represents that the grafted density is achieve the maximum value. To let the modified-NP reach maximum grafted density, based on the Thomson’s theory [34], we adopt a numerical method: a set of points with random velocity and coordinates are constrained on a spherical surface, and they repulse with each other, and then, the average distance of neighbors can be calculated until the system becomes stable. Iterating the process, we can get the maximum number of points when the average distance is equal to the diameter of surficial particles. the maximum number of grafted points is calculated by Equation (1):(1)$$d=<H(r_{i},r_{j})>$$
where *d* is the average distance of neighbor and *r_i_/r_j_* represents particle *i* and its neighbor *j. H* is the nearest neighbor function measured by Euclidean distance. Based on the equation results, we find that when the number of arms is larger than 64, the system will become unstable, therefore, we set this value as η = 1.0. A schematic diagram of PNCs is shown in Figure 1, while Appendix A exhibits the snapshots of PNCs with different η, and Appendix A are shown the root mean square radius of gyration (R_g_) of various modified-NPs.

The typical Lennard-Jones (LJ) potential was employed to model the interaction between different types of beads. The formula of the potential is:(2)$$U_{ij}(r)=\left\{\begin{array}{cc} 4\varepsilon_{ij}\left[{\left(\frac{\sigma }{r-r_{EV}}\right)}^{12}-{\left(\frac{\sigma }{r-r_{EV}}\right)}^{6}\right]-U_{cutoff} & 0<r-r_{EV}<r_{cutoff}\\ 0 & r-r_{EV}>r_{cutoff} \end{array}\right.$$
where *r_cutoff_* denotes the distance at which the interaction was truncated and shifted so that the energy and force were zero. Here the interaction range was offset by *r_EV_* to eliminate the excluded volume effect of different interaction sites. The interaction parameter (*ε*), which endow a measure of the interaction between different type of beads, is shown in Table 1, where the ε between modified-end groups is set as 10.0 to guarantee the strong interaction between the modified-end groups is existed to form the supramolecular crosslink network, and the *ε* between NP-matrix (AA and BB) is set as 3.5 to endows a measure of the interaction between the NP and matrix to ensure the dispersion of NP is uniform [33].

The bond energy between the connected beads in a polymer chain is represented by a harmonic potential:(3)$$U_{bond}=k{\left(r-r_{0}\right)}^{2}$$
the bond is set as $k=100(\varepsilon /\sigma^{2})$, *r*_0_ = 1.12, 2.5, and 1.0, corresponding to the 3 types of bonds: crosslink network, core-NP-grafted chain bead, and other bonds (including grafted polymer chains, matrix AA and BB) [35]. This setting ensured a certain stiffness of the bonds and avoided high-frequency modes and chain crossing. Meanwhile, we build a primary chemical crosslink network to simulate the vulcanization process of rubber nanocomposites in the experiment, ensuring that the rubber nanocomposites possess basic mechanical properties [36]. The chemical cross-linking bonds are generated by a random bonding algorithm, ensuring that all cross-linking bonds will not be generated on the same molecular chain. The number of chemical crosslinking bond is set as 100 [36].

The angle energy between connected beads in a polymer chain is represented by a harmonic potential:(4)$$U_{\mathrm{angle}}=K(\theta -\theta_{0})$$
where *θ*_0_ = 180.0°, and *K* is an alterable parameter influencing the stiffness of grafted polymer chains, and the unit of *K* is set to Rad^−2^.

Because we do not focus on any specific polymer, the reduced LJ units *ε* and σ are used and set to unity, which means that all calculated quantities are dimensionless. In this research, the reduced units of the temperature and the pressure are adopted and defined as follows:(5)$$T^{*}=(k_{B}T/\varepsilon )$$
(6)$$P^{*}=P\cdot \sigma^{3}/\varepsilon$$
where *k_B_* is the Boltzmann constant which is equal to 1 for LJ potential. The simulations have been performed under the NPT ensemble where the temperature is fixed at *T** = 1.0 and *p** = 1.0 by using the Nose-Hoover temperature thermostat and barostat, respectively. Periodic boundary conditions are employed in all three directions. The velocity-Verlet algorithm is used to integrate the equations of motion, with a time step *δ_t_* = 0.001, where the simulation time is represented by the reduced LJ time *τ*. We equilibrate all structures over a long time so that each chain has moved at least 2R_g_, where R_g_ is the root mean square radius of gyration of polymer matrix chains. These fully equilibrated configurations are further used as starting structures for production runs during the structural and dynamic analysis.

The uniaxial tensile deformation has been performed using the protocol implemented in our previous work [37]. The box length along the z direction is increased at a constant engineering strain rate, while the box lengths along the x and y directions are reduced simultaneously, to maintain the constant volume of the simulation box. The engineering strain rate is specified as ${\dot{\varepsilon }}_{1}=(L{\left(t\right)}_{z}-L_{z})/L_{z}=0.0327\tau^{-1}$. The average stress s in the z direction was obtained from the deviated part of the stress tensor: $\sigma_{1}=(1+\mu )(-P_{zz}+P)\approx 3(-P_{zz}+P)/2$, whereas $P=\sum_{i}P_{ii}/3$ was the hydrostatic pressure. The parameter m was Poisson’s ratio, which was equal to 0.5 in the present simulations.

Triaxial tensile deformation is also performed to study the self-healing behavior by following the procedure in our previous work [16]. To achieve triaxial deformation, the simulation box is stretched along the z direction at a constant engineering strain rate, while the box lengths in the x and y directions are kept unchanged. The engineering strain rate is ${\dot{\varepsilon }}_{2}=\frac{\left(L{\left(t\right)}_{z}-L{\left(t_{0}\right)}_{z}\right)}{L{\left(t_{0}\right)}_{z}}=0.015\tau^{-1}$, which was exactly the same as the uniaxial tensile deformation process. The average tensile stress σ in the z direction was obtained from the deviatoric tensor $\sigma_{2}=-P_{zz}$, which is the hydrostatic pressure in the z direction.

All the MD simulations were carried out using the Largescale Atomic/Molecular Massively Parallel Simulator (distributed by Sandia National Laboratories) [38]. More details of the simulation techniques can be found in our previous work [39,40,41,42].

## 3. Results and Discussion

### 3.1. Effect of η on the Structure and Dynamics of PNCs

To characterize the morphology of a supramolecular network with different η values, various types of characterization are implemented. Based on the illustration in Figure 2, we can see that as η increases, numerous modified groups are introduced into the PNCs, causing the supramolecular network to become denser. To quantitatively analyze the formation of the supramolecular network, the number of clusters (N_c_) and the average number of single clusters (N_sc_) are calculated, as shown in Figure 2e. Based on the results, it is observed that as η increases, the size of the clusters decreases. However, when η = 1.0, the size of the cluster increases suddenly. The reason is that when η = 1.0, the excessive concentration of modified groups within the material causes some beads to agglomerate. According to the analysis of the distribution of modified groups (Appendix A), some large clusters (N_sc_ > 90) are formed, while other systems just obtain small clusters (N_sc_ < 10). Therefore, in order to establish a uniform supramolecular network, a moderate number of modified groups are necessary.

To characterize the dispersion of core-NPs, the radial distribution function (RDF) between core-NPs with varying values of η is calculated, and shown in Figure 3b. For η = 0 and η = 0.3, a pronounced peak appears at *r* = 5.14σ, indicating that the NPs are dispersed via segmental-level tight particle bridging. For η = 0.5 and η = 0.8, peaks appear at *r* = 5.75σ and 7.87σ, indicating that there are two states of NP dispersion: segmental-level tight particle bridging and adsorbed layers coexisting with longer range bridging. For η = 1.0, only one peak appears at *r* = 7.92σ, indicating that the core-NPs are sandwiched by grafted chains and polymer matrix. Meanwhile, based on the results in Figure 3b, when 0 < η < 1, the maximum value of *g*(*r*) is reduced as η increases, signaling an improvement in the dispersion of core-NPs. In general, the characterization of RDF proves that the dispersion of NPs is governed by the configuration of the supramolecular network.

The effect of η on T_g_ is investigated and the results are shown in Figure 4a, which were obtained where we identified the temperature at which the specific volume varied. The curves of specific volume-temperature are shown in Appendix A. It is found that as η increases, the value of T_g_ increases synchronously and slowly, indicating that the completeness of the supramolecular network affects the mobility of polymer chains. This suggests that the dynamic crosslink network creates a hindrance effect, which hinders the mobility of polymer chains.

To quantitatively analyse the relationship between η and the mobility of polymer chains, the mean-square displacement (MSD) is calculated, which is shown in Figure 4b,c. As η increases, the value of MSD decreases obviously, which proves the inference summarized in Figure 4a. Meanwhile, the bond autocorrelation function, *C_b_*(*t*), as a function of η, is calculated to further measure the dynamics of polymer chains. The equation of *C_b_*(*t*) is exhibited as follows:(7)$$C_{b}(t)=(\mu (t)\times \mu (t_{0}))$$
where *μ(t)* denotes a unit vector characterizing the orientation at a time. Figure 5 shown after a prolonged relaxation process, the decay of *C_b_*(*t*) decreases in tandem with an increase in η, signifying the significant impact of the supramolecular network on the relaxation behavior of polymer chains. Furthermore, it is observed that when η > 0.5, the minimum value of *C_b_*(*t*) is higher than 1/e, which further proves the negative correlation between the completeness of the supramolecular network and the relaxation of polymer chains.

The entanglement of the polymer chains was characterized using the Z1 code to determine the confinement effect of η on segment dynamics [43,44,45,46]. The entanglement network analysis for different PNCs is summarized in Table 2. Based on the results, the increase of η corresponds positively to the mean-squared end-to-end distance $\sqrt{R^{2}}$, indicating that the modified NPs absorb the polymer chains on the surface. The value of the average number of entanglements per chain, denoted by <*Z*>, is also enhanced with an increase in η, signifying that the formation of the supramolecular network significantly affects the entanglement network.

Meanwhile, it is observed that although the introduction of modified groups accelerates the entanglement behavior of polymer chains, when η > 0, the value of the mean contour length of the polymer chains (*L_p_*) decreases and exhibits an irregular tendency with an increase in η. This suggests that the supramolecular network affects the flexibility of the polymer chains, and the effect of η on the flexibility of polymer chains is unpredictable. This unpredictability may affect the process ability of PNCs.

### 3.2. The Static Mechanical Performance and Self-Healing Behavior of PNCs

To investigate the influence of η on the mechanical performance of PNCs, uniaxial deformation tests of PNCs were carried out. As η increases, the tensile stress of PNCs increases synchronously, indicating that the supramolecular network reinforces the mechanical performance of PNCs as shown in Figure 6a. It is noteworthy that when η = 1.0, the curve breaks when the tensile strain is equal to 5, indicating that excessive dynamic crosslink nodes can affect the mechanical performance of PNCs. Except for this special case, when η = 0.8, the PNC exhibits the best static mechanical property. The chain orientation behavior during uniaxial deformation process is also examined by employing the second-order Legendre polynomials, which is exhibited as follows [33,47]:(8)$$<P_{2}(\cos\theta )>=\frac{1}{2}[3(\cos^{2}\theta )-1]$$
where *θ* denotes the angle between a given element (two adjoining monomers in the chains) and the reference stretching direction. The possible values of $<P_{2}(\cos\theta )>$ range from −0.5 to 1, and the values of −0.5, 1 and 0 indicate a perfection orientation perpendicular to the reference direction, a perfection orientation parallel to the reference direction or randomly oriented, respectively. Based on the results in Figure 6b, the value of $<P_{2}(\cos\theta )>$ is negatively correlated with η. Especially when the tensile strain is relatively large (>5), the difference in $<P_{2}(\cos\theta )>$ is apparent. This result illustrates that the supramolecular network highly affects the orientation behavior of polymer chains, and the major contribution to the mechanical performance of PNCs belongs to the supramolecular network, not the reinforcement of NPs or polymer matrix chains.

To understand the self-healing behavior of PNCs, the stress-strain curve of PNCs with various values of η was studied as a function of self-healing temperature under triaxial deformation. Firstly, the 1st triaxial deformation process was carried out, which is shown in Figure 7a. Based on the results, it was observed that when the tensile strain is relatively small (<1), the tensile stress is positively correlated with the increment of η. This indicates that the completeness of the supramolecular network enhances the mechanical performance of PNCs, which is consistent with the conclusion obtained from Figure 6. However, as the triaxial deformation is conducted, the system with a high η exhibits low tensile stress. This means that when the crosslink network is broken, the number of dynamic crosslink nodes is not necessary for the mechanical performance of PNCs.

Then the influence of temperature on the self-healing efficiency, *ψ_sh_*, of PNCs is studied. *ψ_sh_* is defined as the σ_max,z_ (the maximum tensile stress along z direction with different self-healing temperature in 2nd triaxial deformation) over σ_max,o_ (the maximum tensile stress at 1st triaxial deformation). The PNCs were deformed by the triaxial deformation and compressed afterward. Then, they were kept in a range of fixed temperatures for a period of time. Based on the results in Figure 4a, all systems obtain T_g_ lower than 1.0, so the self-healing temperatures are set as 1.0, 1.5, 2.0, 2.5 to study the self-healing behavior of PNC, which ensure the PNCs are in rubbery state. Based on the results in Figure 8, it is observed that the value of *ψsh* is positively correlated with the temperature, signaling that the increase in temperature enhances the self-healing process of PNCs. It is observed that when η = 0.3, the value of *ψsh* is decreased. This phenomenon indicates that low content of modified-end groups cannot enhance the re-establishment of supramolecular crosslink network, but hinder the re-build process of crosslink network. Therefore, it is necessary to ensure the moderate modified-end groups to accelerate the self-healing behavior of PNCs. Meanwhile, it can be seen that for η ≥ 0.5, when T = 2.0 and 2.5, the difference in *ψ_sh_* is not distinct, indicating that the self-healing efficiency has a limitation value dependent on the self-healing temperature when the completeness of supramolecular crosslink network is relatively high. Meanwhile, as the increment of η, the efficiency is generally enhanced, indicating that the improvement of the supramolecular network benefits the self-healing efficiency. Nonetheless, when η > 0.8, the value of *ψ_sh_* stops increasing, and even decreases slightly, indicating that the excessive modified groups are a shortcoming of the self-healing behavior. This inference proves that the supramolecular network with moderate completeness is the most suitable selection for designing self-healing PNCs.

To investigate the morphology of PNCs during triaxial deformation, we examine the contribution of each component to the total stress, σ_T,_ during two triaxial deformations. We evaluate the value of σ_T_ using color, where redder denotes higher σ_T_ and bluer denotes lower σ_T_. To make the argument more representative, we consider systems with η values of 0, 0.5, and 1.0, which represent low, medium, and high completeness of the supramolecular network, respectively. As shown in Figure 9, we observe that as η increases, the stress distribution becomes narrower during either the 1st or 2nd triaxial deformation, which confirms the analysis based on Figure 7a. Furthermore, when the completeness of the supramolecular network is low or medium, the structure of the PNC does not fully collapse during the 2nd triaxial deformation. Meanwhile, it is observed that when η = 1.0, the stress distribution becomes narrower than η = 0.5, and still contain the crosslink structure stable under high self-healing temperature, while the structure is totally collapse under low self-healing temperature, signaling that for the relatively high grafted density, the high temperature is a necessary condition to achieve remarkable self-healing efficiency.

## 4. Conclusions

In this study, the effect of the structure of modified-NPs on the performance of PNCs is investigated through CGMDs. A series of PNCs with different sphericities (η) is established. First, the morphology of the supramolecular network and the dispersion of NPs are characterized. It is observed that as the η increases, the dispersion state of the NP is transferred from particle bridging to being sandwiched by polymer chains, and the increment of η enhances the completeness of the supramolecular network. Furthermore, the investigation of the segment’s dynamic shows that the mobility of the polymer chains was highly dependent on η, while the increment of η has a negative correlation with the mobility and relaxation behavior of polymer chains. The variation of the average number of entanglements per chain, <*Z*>, indicates that the completeness of the supramolecular network reinforces the entanglements of polymer chains significantly. The mechanical performance of the PNCs indicates that the contribution of the mechanical properties is mainly from the supramolecular network, and fully modified-NPs will damage the static mechanical performance of PNCs. The self-healing behavior of PNCs with different η is also investigated. The results show that the η has a positive correlation with the self-healing efficiency, and high temperature can improve efficiency as well. However, based on the study of the stress heatmap of PNCs, it is observed that high η will make the structure of PNCs unstable, which is against the stability of the supramolecular network. Furthermore, it is proven that the inference from the analysis of the mechanical responses of PNCs during triaxial deformation. This study offers valuable insights into the impact of modified NP structure on the self-healing behavior of PNCs. Furthermore, this study proposes a novel perspective on the relationship between the integrity of the dynamic crosslink network and the self-healing performance of PNCs.

## Figures and Tables

**Figure 1 polymers-15-03259-f001:**
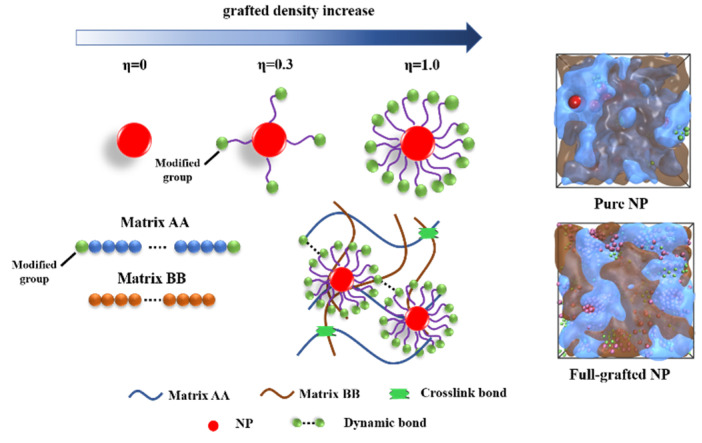
The schematic diagram of self-healing PNCs. The red spheres represent the core-NP, the purple lines represent the grafted polymer chains with length = 4σ, the green spheres represent the modified groups that provide a strong interaction strength to establish the dynamic crosslink network. The blue balls represent the matrix AA and the brown balls represent the matrix BB.

**Figure 2 polymers-15-03259-f002:**
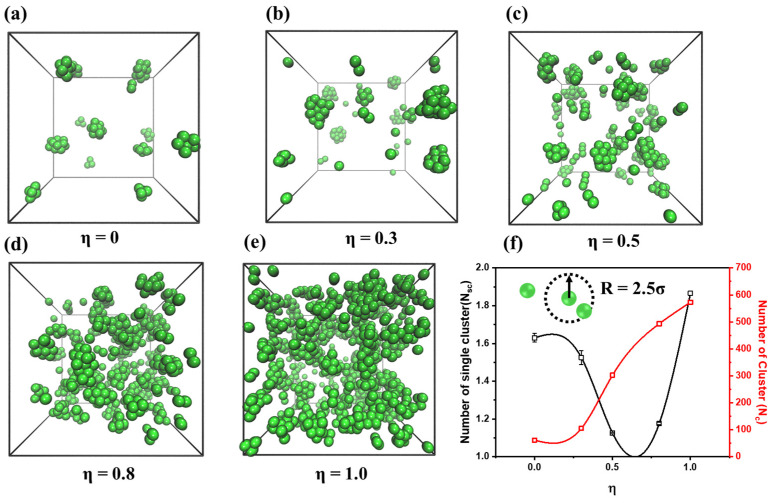
The snapshot of modified group that are formed the supramolecular network, where: (**a**) η = 0; (**b**) η = 0.3; (**c**) η = 0.5; (**d**) η = 0.8; (**e**) η = 1.0. The green balls represent the modified-end groups of PNCs. (**f**) the number of clusters (black line, N_c_) and the average number of single clusters (red line, N_sc_) as a function of the variation of η. The black arrow represents the definition radius of cluster.

**Figure 3 polymers-15-03259-f003:**
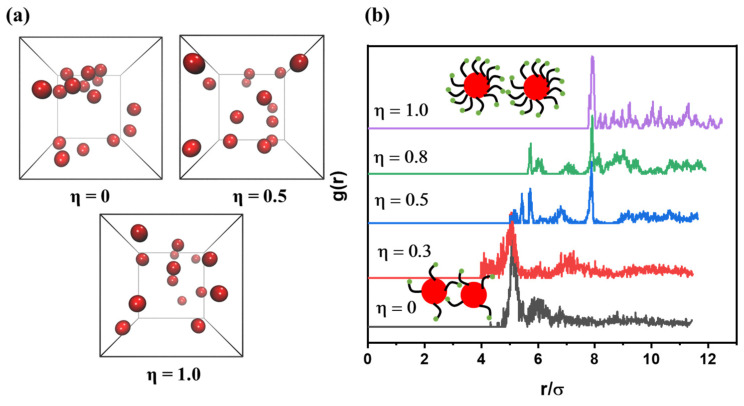
(**a**) the snapshot of the dispersion of core-NPs when η = 0, 0.5 and 1.0, (**b**) the radial distribution function (RDF) between core-NPs as a function of η.

**Figure 4 polymers-15-03259-f004:**
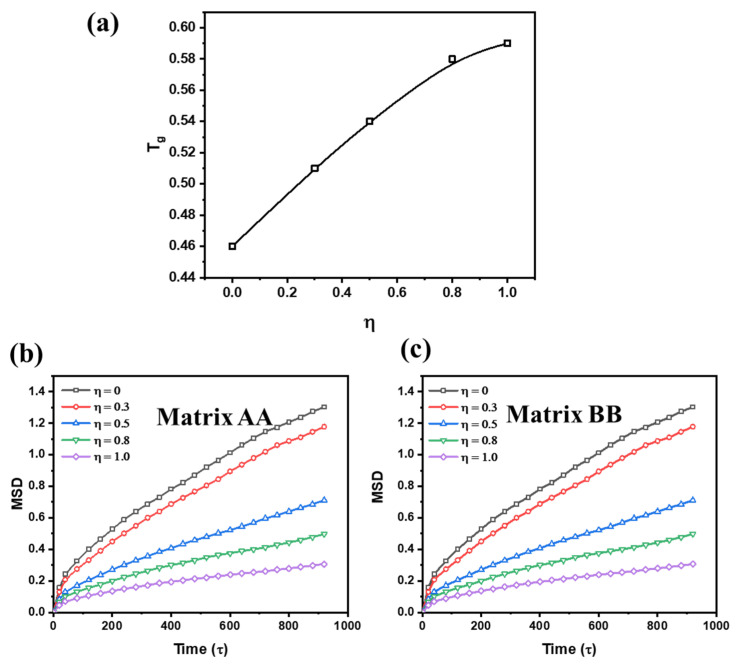
(**a**) the variation of T_g_ with the increment of η, and the mean-square displacement (MSD) as a function of time with different value of η, where (**b**) is MSD of matrix AA and (**c**) is MSD of matrix BB.

**Figure 5 polymers-15-03259-f005:**
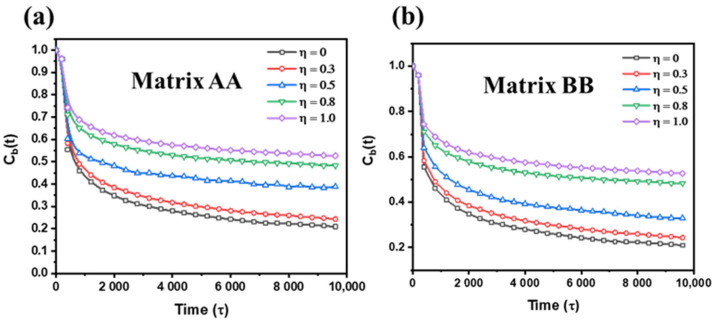
The bond-autocorrelation function *C_b_*(*t*) as a function of simulation time with various η, where (**a**) is matrix AA and (**b**) is matrix BB.

**Figure 6 polymers-15-03259-f006:**
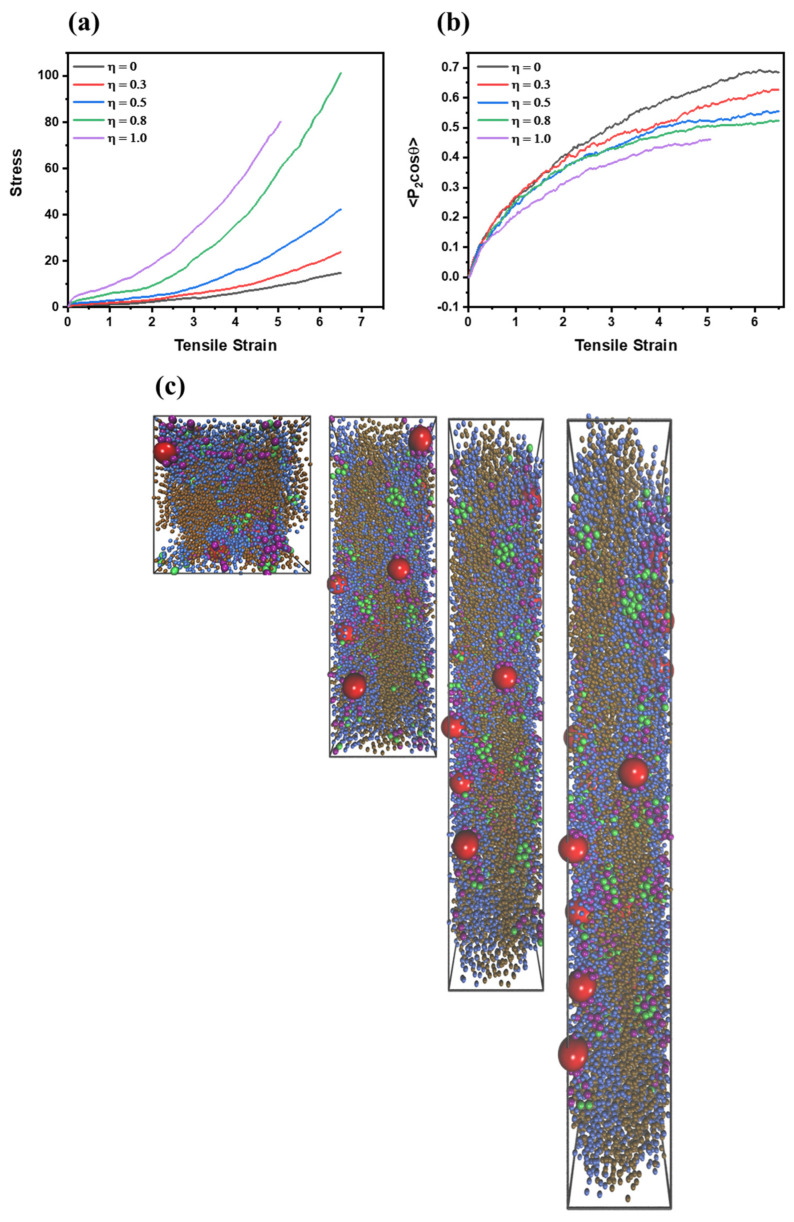
(**a**) the stress-strain curve of PNCs and (**b**) the chain orientation of polymer matrix (including AA and BB) chains with various η, (**c**) the snapshot of PNCs when η = 0.5 with different tensile strain during uniaxial deformation.

**Figure 7 polymers-15-03259-f007:**
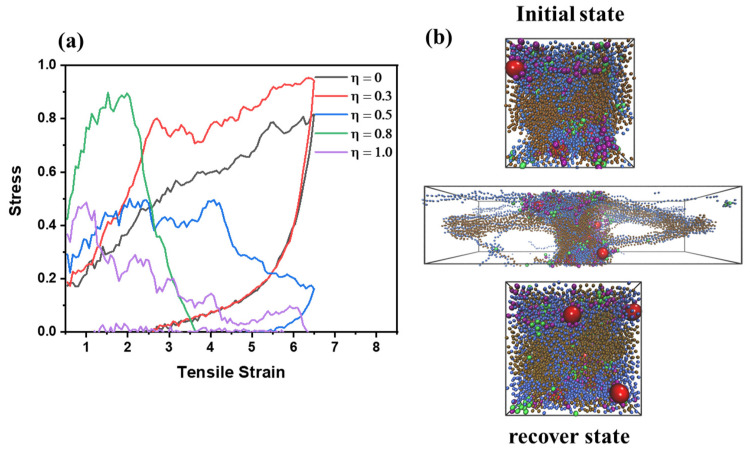
(**a**) the stress-strain curve of PNCs with different η. (**b**) the snapshot of PNCs when η = 0.5 with different tensile strain during triaxial deformation.

**Figure 8 polymers-15-03259-f008:**
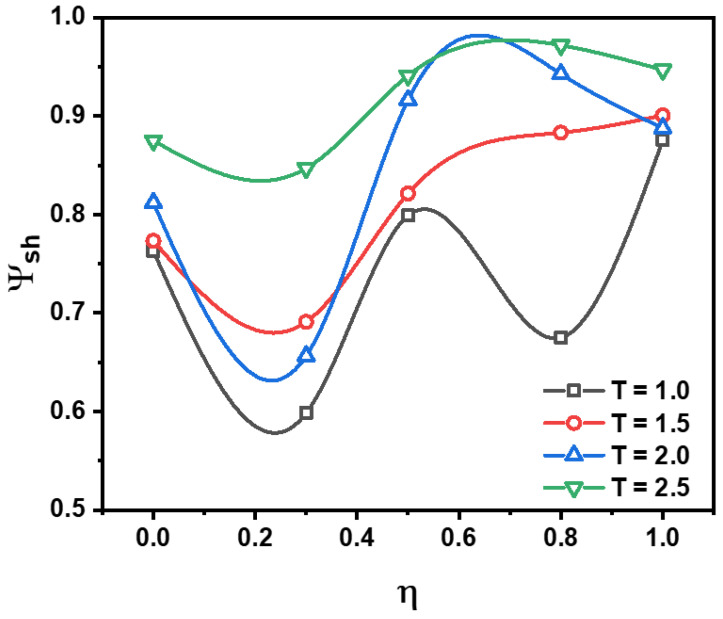
The self-healing efficiency (*ψ_sh_*) of PNCs with various η as a function of self-healing temperature, where *ψ_sh_* is defined as the σ_max,z_ (the maximum tensile stress along z direction at different self-healing temperature) over σ_max,o_ (the maximum tensile stress at 1st triaxial deformation).

**Figure 9 polymers-15-03259-f009:**
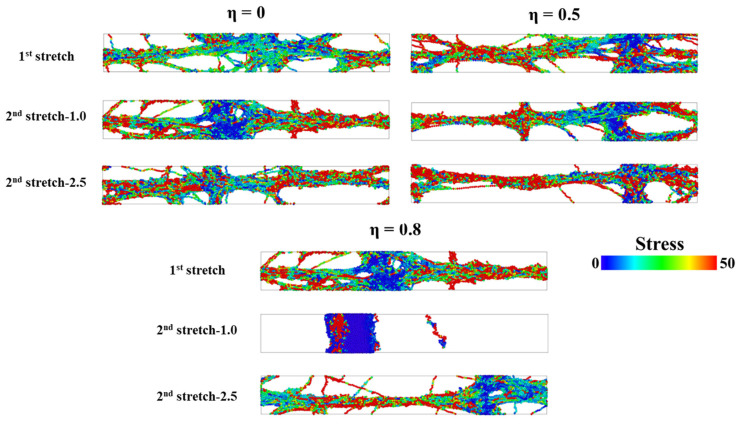
The heatmap of total stress (σ_T_) of PNCs with various η and temperature during 1st and 2nd triaxial deformation. For all systems, the tensile strain is set as 3.

**Table 1 polymers-15-03259-t001:** The interaction parameter setting for different type of beads.

Interaction Type	Interaction Parameter (*ε*) Value	Cutoff Distance *r_cutoff_*
Modified group-Modified group	10.0	2.5
NP-matrix (AA and BB)	3.5	2.24
NP-grafted beads	1.0	1.12
others	1.0	2.5

**Table 2 polymers-15-03259-t002:** The entanglement network analysis of PNCs with various η.

System	$\sqrt{R^{2}}$	*L_p_*	*<Z>*
η = 0	12.158	17.384	2.950
η = 0.3	89.174	11.098	17.404
η = 0.5	95.435	11.339	20.276
η = 0.8	103.783	14.404	33.281
η = 1.0	120.478	14.82	41.101

## Data Availability

No new data were created or analyzed in this study. Data sharing is not applicable to this article.

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
