# Peer review of "Molecular Dynamics Simulation of Polymer Nanocomposites with Supramolecular Network Constructed via Functionalized Polymer End-Grafted Nanoparticles"

_polymers, 2023, doi:10.3390/polym15153259_

Round 1
Reviewer 1 Report
The research conducted by the authors focused on exploring the relationship between the morphology of the dynamic crosslink network and the comprehensive properties of self-healing PNCs. While there have been previous studies on self-healing materials, this particular aspect had received limited attention, making this research valuable and significant. The paper can be accepted after major revision. Some important issues should be clarified.
Understanding the type of self-healing polymer nanocomposites reviewed in this paper is challenging based on the introduction alone. It would be beneficial to provide some clarification or context regarding the specific materials under investigation. Additionally, it is important to highlight that the self-healing process often requires external stimuli to enhance the rate of the chemical reaction, as it typically occurs at a low rate under room conditions.
The authors discuss hypothetical dynamic bonds and the formation of cross-links, but it is crucial to address the variations in reactivity, inactivation, and the slow rate of reaction that exist among different functional groups in real-life scenarios. This information should be included and explained in the paper to provide a more comprehensive understanding of the topic. Furthermore, it would be helpful to suggest the interaction of specific functional groups in practical applications.
The authors briefly mention the influence of temperature on self-healing efficiency, specifically in the context of the glass transition temperature (Tg). However, if I understood correctly, the self-healing mechanism in this system is based on a different approach. It is essential to provide a clear explanation of this viewpoint to avoid confusion and to help readers grasp the key aspects of the research.
The concept introduced in the paper referred to as "sphericity (η)," and its relation to the grafted density of nanoparticles (NPs) needs to be clarified. It is necessary to explain the practical implications of considering grafting density, which theoretically can exceed 1 depending on the amount of grafted molecules and its physical significance within the context of the study. Providing additional information and clarification in this regard would enhance the reader's understanding.
Finally, I suggest citing the paper where similar systems were reviewed: https://doi.org/10.1007/s00396-020-04750-0
Minor editing of English language required.
Reviewer 2 Report
My comments for the manuscript entitled, ‘Molecular dynamics simulation of polymer nanocomposites with supramolecular network constructed via functionalized polymer end-grafted nanoparticles’ are given below,
1. As could be seen, the main objective of the work is to relate morphology of the network with their self-healing properties. But morphology of the polymer nanocomposites is not discussed anywhere in the manuscript.
2. Generally, addition of nanoparticles to polymer matrix increases their tensile strength. But a reverse behavior is observed in the present work. Reason it out?
3. When η value is small, it indicates smaller particles, and so there are more chances of agglomeration than particles with high η value. But it is mentioned that particles with high η value gets agglomerated. Explanation is needed in this regard.
4. Figure 4b and c looks similar. Recheck it.
5. In Figure 9, include the heat map for η=0.3 and 0.8. Moreover η=0.8 shows better property and so its heat map is necessary for comparison.
6. It is mentioned that ‘The reason is that when η = 1.0, the excessive concentration of modified groups within the material causes some beads to agglomerate’, but actually with larger size there will be less amount of agglomeration. The reason for decreased property should be due to the less amount of particles in the composites.
7. Self-healing property decreases with increased size of the nanoparticles and also when compared with η=0. Explanation is needed in this regard.
After rectifying these comments, the manuscript can be considered for publication
Round 2
Reviewer 1 Report
The authors have answered all my comments and the paper can be accepted to publication in its present form.
Minor editing of English language required.
Reviewer 2 Report
The authors have answered all my questions, and the manuscript is now acceptable.
Minor editing of English language required
Round 3
Reviewer 2 Report
The manuscript in the present form can be accepted
English language fine. No issues detected